# Assessing the efficacy of a job rotation for improving occupational physical and psychosocial work environment, musculoskeletal health, social equality, production quality and resilience at a commercial laundromat: protocol for a longitudinal case study

Jennie A. Jackson ![ORCID],[1] Marianne Sund,[2] Griztko Barlari Lobos,[2] Lars Melin,[2] Svend Erik Mathiassen[1]

¹Department of Occupational Health Sciences and Psychology, University of Gävle, Gavle, Sweden
²Elis Textil Service AB, Ockelbo, Sweden

**Correspondence to**
Dr Jennie A. Jackson;
jennie.jackson@hig.se

## ABSTRACT

**Introduction** Job rotation is a work organisation strategy used to reduce work-related exposures and musculoskeletal complaints, yet evidence for the efficacy of the approach is weak. Mismatch between job rotation and company needs, lack of full implementation, lack of exposure variation in included tasks and failure to assess variation may underlie inconclusive research findings to date. The study aims to develop a job rotation with company stakeholders, perform a process evaluation of the implementation, and determine the extent to which the intervention improves the physical and psychosocial work environment, indicators of health, gender and social equality among workers and production quality and resilience.

**Methods and analysis** Approximately 60 production workers at a Swedish commercial laundromat will be recruited. Physical and psychosocial work environment conditions, health, productivity and gender and social equality will be assessed pre and post intervention, using surveys, accelerometers, heart rate, electromyography and focus groups. A task-based exposure matrix will be constructed, and exposure variation estimated at the level of the individual worker pre and post intervention. An implementation process evaluation will be conducted. Job rotation efficacy will be assessed in terms of improvement in work environment conditions, health, gender and social inequality, and production quality and resilience. This study will provide novel information on the effects of the job rotation on physical and psychosocial work environment conditions, production quality and rate, health and gender and social inequality among blue-collar workers in a highly multicultural workplace.

**Ethics and dissemination** The study received approval from the Swedish Ethical Review Authority (reference number 2019-00228). The results of the project will be shared directly with the employees, managers and union representatives from the participating company, other relevant labour market stakeholders and with researchers at national and international conferences and via scientific publication.

**Trial registration number** The study is preregistered with the Open Science Framework (https://osf.io/zmdc8/).

## STRENGTHS AND LIMITATIONS OF THIS STUDY

⇒ Cocreation participative intervention and study design.
⇒ Assessment of job rotation intervention effects on physical and psychosocial work environment conditions, health, production quality and resilience with consideration to gender and social equality.
⇒ Process evaluation of the intervention implementation.
⇒ Consideration of inequality both from a male–female (gender) and a place of birth (ethnicity).
⇒ Case study based on a limited number of workers at a single work site.

## INTRODUCTION

Job rotation is an organisation-level strategy that involves alternating workers between tasks that, ideally, differ in physical and psychosocial demands. In the field of ergonomics, job rotation has predominantly been used with the aim of reducing work-related exposures and musculoskeletal complaints; however, the current evidence has been described as weak[1] or inconclusive[2] regarding the efficacy of job rotation to prevent musculoskeletal disorders (MSDs). Recent work has even suggested that job rotation can increase the overall risk of workplace injury.[3]

The efficacy of a job rotation is theoretically dependent on the extent to which the

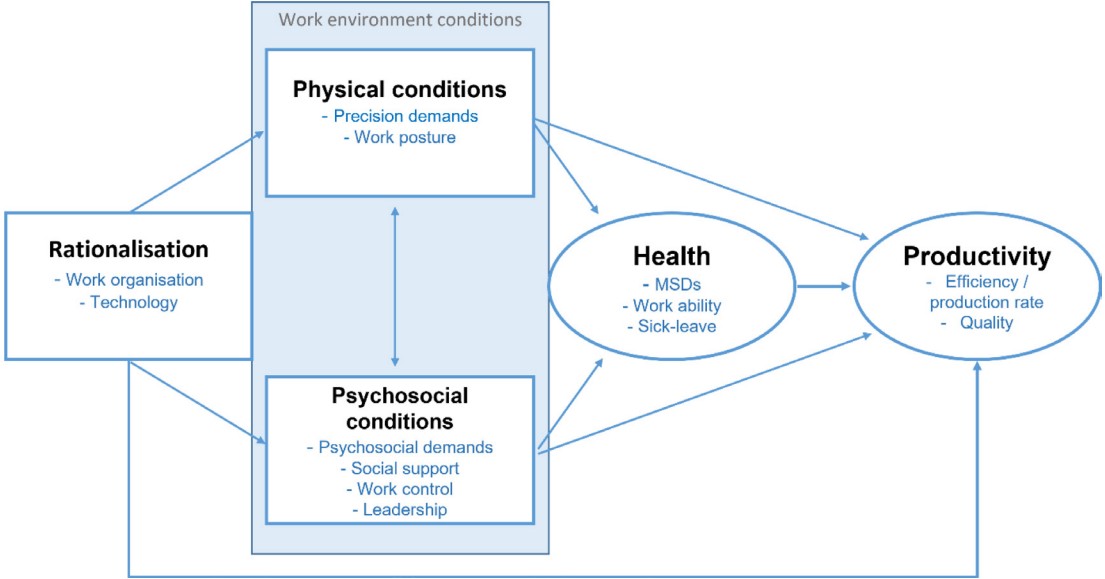

**Figure 1** Conceptual model illustrating overlapping interests of production and ergonomics. Proposed interactions are shown in arrows between rationalisation (job rotation, in the current study) and work environment factors (shown in rectangles), and health and productivity outcomes (shown in ovals). MSDs, musculoskeletal disorders. Model adapted from Winkel and Westgaard[11] and Rolander et al.[12]

tasks included in the rotation differ, and thus the extent to which more variation can be introduced in the jobs of individual workers rather than appearing between workers.[4] Quantification of exposure variation is therefore required before and after job rotation to determine whether the intervention succeeded in altering exposure patterns. Mixed study results to date regarding the effects of job rotation on health outcomes may be the result of a lack of objective assessment of exposure variation both before and after intervention. In Leider's 2015 systematic review of evidence of the effects of job rotation on musculoskeletal complaints, only 4 of the 16 studies quantified aspects of both exposure and MSD health outcomes, and none of the studies assessed exposure variation.[5–8]

Consideration of psychosocial factors is also paramount when considering the complex relationships between work organisation, work environment and health (including MSDs)[9–13] (figure 1). Similar to physical exposures, job rotation is generally undertaken to improve psychosocial conditions. While positive correlations have been shown for higher worker job satisfaction ratings,[1] there is inconclusive evidence to date for the effects of job rotation on other psychosocial risk factors, for example, job control[14] or health outcomes, including work absence or sick leave.[1] Again, mixed study results to date may be the result of low-quality study designs and the lack of assessment of exposure and outcome variables before and after intervention.

The conceptual model used in this study is given in figure 1 and illustrates the interconnection of ergonomic and production factors and outcomes which warrant consideration when making changes in work organisation. Rationalisation is defined here as comprising changes made to 'maximise productivity, quality and efficiency

under the prevailing conditions of, among other things legislation, education level of available work force and culture'.[11] In addition to potential ergonomic gains, job rotation has been shown to improve employee skill/competence, productivity, quality,[15 16] decrease worker turnover[15] and improve organisational flexibility.[16] Thus, job rotation is per se a rationalisation. At the time this paper was written, we were unable to find any studies that had simultaneously and quantitatively addressed the effects of a job rotation on the interrelated ergonomics and production factors and outcomes outlined in figure 1. The job rotation will be designed according to the Goldilocks Work approach[17 18] to distribute physical and mental loads during productive work to improve worker health and, to the extent possible, capacity. The job rotation will aim to distribute tasks with higher physical and/or mental demands across workers, and to schedule work to facilitate recovery by interspacing higher demand tasks with lower demand tasks.

Inconclusive findings in previous job rotation studies may also be the result of a mismatch between specific company needs and the implemented job rotation.[19 20] To improve intervention efficiency, a participatory approach has been called for in which interventions are cocreated by researchers together with relevant company stakeholders, such as, management, health and safety, union, and production representatives.[21 22] Cocreation can also increase knowledge within an organisation regarding the problem(s) to be targeted by an intervention and how the intervention will address the problem(s). This can enhance readiness for change, which is a necessity for successful intervention implementation.[23 24] Cocreated programme logic (COP) is a participatory approach that begins with a set of outcomes identified by stakeholders,

around which a specific intervention is subsequently designed.[25] COP is also a tool for defining, through negotiation between stakeholders and researchers, the outcomes to be measured and the optimal time points for the measurements. COP has proved successful in developing study aims and design and in improving readiness for change, intervention contextual fit and implementation efficacy.[23]

Insufficient process evaluation of the intervention implementation, including failure to fully implement the intervention,[26] has also been identified as a common weakness in prior job rotation studies.[27] Consideration of how job rotation has been implemented in subpopulations of workers may prove salient in fully understanding the implementation and in the effects of an intervention, but to date has been sparse. Findings from a job rotation intervention study at a grocery store found work tasks deemed the worst by employees were excluded from the intervention and subsequently allotted to a group of female workers who were, thus, not included in the job rotation.[28] This example is in line with the findings of a 2013 review report prepared for the Swedish Work Environment Authority, finding that task allocation within jobs was gendered, where female workers tended to be given more monotonous and repetitive tasks than male workers.[29] Further, there is a broad consensus in the literature that women receive less on-the-job training than men, including initial on-the-job training,[30] which positions women to be assigned fewer tasks and possibly to be excluded for job rotation initiatives comprising tasks requiring more than basic training or competences. Foreign-born employees have also been shown to receive less on-the-job training than native born employees.[31] Consideration of job rotation design and implementation within a social context may therefore be crucial. The many limitations of previous studies and the lack of research simultaneously considering aspects of social inequality in task assignment, job rotation and health demonstrate the need for further research that considers the effects of job rotation from a wide range of perspectives.

The aim of this study is to develop a job rotation intervention at a commercial laundromat in concert with company stakeholders using COP, evaluate the process implementation of the job rotation intervention and determine the extent to which the intervention can improve the physical and psychosocial work environment and indicators of health, advance gender and social equality among workers and increase production quality and resilience without negatively affecting production rates.

The specific aims of the study are to:

1. Describe the cocreation programme logic process used to develop the intervention and study design.
2. Provide a process evaluation for the implementation, including assessment of readiness for change and the extent to which the intervention was implemented.
3. Determine the extent to which the job rotation intervention increased exposure variation and lead to changes in physical and mental work environment factors and indicators of health.
4. Evaluate social and gender inequality in work organisation, working conditions and health before, during and after the job rotation intervention.
5. Assess the impact of the job rotation on production rate, quality and resilience.

## METHODS AND ANALYSIS

The study will be performed at a commercial laundromat employing approximately 60 full-time workers in laundry handling tasks. The study resulted from contact initiated by the study site: a commercial laundromat in Sweden. The laundromat management and union saw room for improvement in physical and psychosocial working conditions and issues of equality and were interested in implementing a job rotation, but sought help to successfully make the change. We began by establishing a steering committee of eight people that comprised company management (LM and GBL), production workers, health and safety and union representatives (MS) and researchers (JJ and SM) to jointly develop the study.

After extensive negotiations, the steering committee arrived at a cocreated study design that included (1) development of an innovative job rotation intervention via cocreation with company stakeholders, (2) evaluation of the implementation of the job rotation intervention and (3) assessment of the efficacy of the job rotation intervention using a pre–post intervention case study design. Methods for all three aspects of the study are presented below.

### Development of the job rotation intervention using COP

The job rotation intervention was cocreated in a collaboration between management (GBL), union representatives (MS) and researchers (JJ) and was guided by the Goldilocks Work approach principals[17 18] to find a health-promoting balance in both physical and mental work exposures. To better understand laundry processing tasks, one researcher (JJ) shadowed workers performing each task and, to the extent possible, spent 15–45 min performing each task. Company representatives and the researcher worked together to rate the physical and mental load (heavy/light) and the degree of repetitiveness (monotonous/variable) of each task. Led by the research team, company representatives grouped tasks into different proficiency levels, starting with a base group of tasks that they believed all workers could and should learn and then moving onto tasks that required increased knowledge of production flow and task-specific training. Five hierarchical levels of tasks were created, namely: (1) basic tasks (which all workers could perform), (2) driers and small machines, (3) order picking and packing for hotel and medical customers, (4) order picking and packing of work clothes and (5) steering production flow.

The laundromat is physically divided into two sides—the sorting side, where bags of dirty laundry are received,

emptied onto conveyor belts and goods are sorted by colour and item prior to washing, and the processing side of the laundromat, where clean goods are dried, pressed, folded and packed. In general, rotation from sorting to processing is not typically done within a day, as a shower is required before moving from sorting to processing. Base tasks were identified on both the sorting and processing sides, and so full-day rotation plans were established for each side to accommodate the hygiene requirements.

To meet company goals for improved language proficiency, social and gender equality and integration, the job rotation was based on teams comprising 3–6 workers that rotated as a unit. Teams were created to include at least one man and one woman, one native Swede, ideally not more than one person with the same mother tongue and a mix of seniority levels in the competencies required for the tasks assigned to the team. A total of 10 job rotation teams were formed; 5 of which performed basic tasks. Of these, four teams were assigned both sorting work and processing work, while one base group was assigned only processing work. Job training was required for nearly all workers to be able to perform all tasks assigned under the job rotation intervention. The company aimed to, in parallel, develop a new salary structure that would match the new proficiency 'steps' and competencies with employee wages. The company expected that the new work and wage structure would make for a more transparent development and promotion path, including making it easier for employees to understand and express interest in career advancement plans.

### Implementation process and outcome evaluation

To assess the implementation process, semistructured focus group interviews with each job rotation team and individual interviews with management will be conducted at three time points: immediately prior to the start of the job rotation intervention and at 6-week and 12-month follow-up. Data collected immediately prior to the job rotation intervention will be used to assess worker readiness for change and worker impressions of their new working schedules. Individual interviews with managers

from this time point will be used to assess readiness to lead change. At 6-week follow-up, group interviews will be used to document the extent to which the job rotation intervention has been implemented during the initial phase and self-reported ability of workers to complete all assigned tasks. At 12-month follow-up, group interviews with each team will be used to document the progress and extent of implementation achieved. Individual interviews with management will be used to assess the company's perspective and experience of the implementation process and the extent of success. Together, the interviews will capture facilitators and barriers to the implementation process and will document additional changes occurring during the follow-up period that were unrelated to the intervention, but which may have affected the way in which work was performed.

Interview data will be transcribed and analysed thematically,[32] guided by research aim 2 to assess readiness for change and the extent of intervention implementation.

### Assessment of job rotation intervention efficacy via pre–post study design

Using COP[25] as earlier outlined, outcome metrics were established by the researchers in concert with the company steering committee using an iterative procedure based on identifying desired changes that could directly be tied to the job rotation intervention. The study scientific testing protocol was then developed (JJ and SM) and presented to the steering committee who completed a risk analysis prior to the protocol being approved.

### Study design overview

The workers will be followed for approximately 12 months, with measurements occurring prior to, during and 8–12 months following the job rotation intervention implementation (figure 2). Baseline physical and psychosocial working conditions will be documented using (1) a questionnaire to assess the physical and psychosocial environments, (2) technical measurements to assess posture, heart rate (HR) and muscle activity levels during work and (3) semistructured interviews to assess readiness

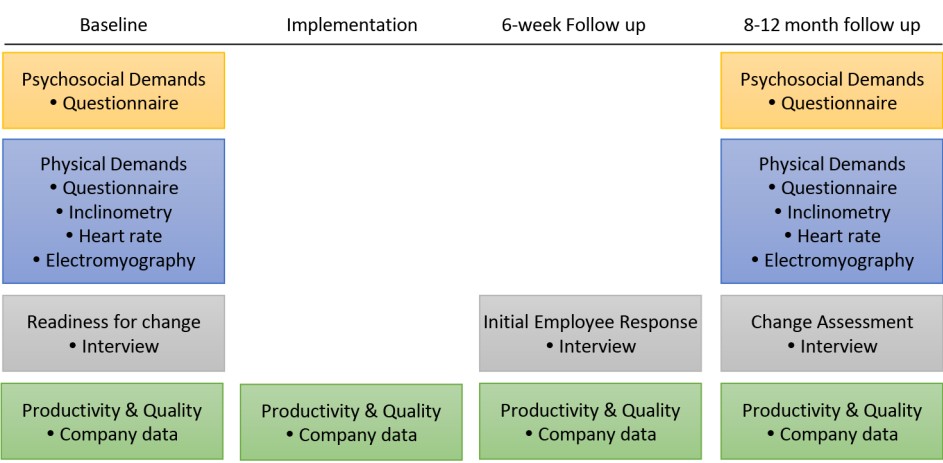

**Figure 2** Study design.

for change. Initial response to the intervention will be assessed 6 weeks post implementation via semistructured focus group interviews. At follow-up, physical and psychosocial working conditions will be reassessed using the baseline measurement protocol, and a final set of semistructured focus group interviews will be conducted to document production and working environment changes that occurred during the follow-up period but were unrelated to the job rotation intervention.[33] Production rate and quality will be assessed weekly throughout the study period, using data collected by the company.

## Study population

Permanent and contract workers currently employed in laundry handling tasks at the laundromat will be invited to participate in the study; the exact number of participants and inclusion criteria will vary by measurement type and research question, as outlined below. For the questionnaire, we will aim for a response rate of 70% (n=35). For both technical measurement protocols, we will collect data from all interested participants and will aim for a minimum of 20 participants, which corresponds to approximately 30% of the total worker group. Focus groups interviews will be conducted with each team and will include all team members who elect to participate.

Approximately half of the employees are foreign born with varying levels of Swedish language proficiency, and there is a wide range in education levels across all employees. To ensure workers are informed, understand participation is voluntary and understand what participation entails, we will present all aspects of the research project in person to all employees using slides relying primarily on graphical displays of information, and we will provide live demonstration of technical measurement methods.

All production employees will be invited to participate in the study. Baseline survey response rates were 97% (n=58), which suggests a strong likelihood for obtaining ample data for this case study throughout the follow-up period. Focus on developing and maintaining a strong relationship with the company has been present since study inception.

### Pre and post job rotation measurements
#### Questionnaire

A custom questionnaire comprised of modules from well-established, validated and documented questionnaires and modules adapted from questionnaires previously used in our research group to assess other occupational groups, but adjusted to specifically capture information about the commercial laundromat environment.[34 35] The modules in our questionnaire consider: personal and demographic parameters (9 questions), employment status and working hours (7 questions), specific work tasks currently performed and time spent per task over a week (6 questions), physical and mental demands of each work task and the overall job (4 questions with Borg category-ratio (CR) 10 response scale[36]), self-rated work ability (4 questions), workload, fatigue and recovery (short Swedish Occupational Fatigue Inventory (SOFI) Questionnaire,[37 38] 14 questions); musculoskeletal pain (Standardised Nordic Questionnaire,[39] 4 questions), job satisfaction (2 questions) and the psychosocial work environment (selected Copenhagen Psychosocial Questionnaire (COPSOQ) modules,[40] 79 questions).

All employees will be invited to complete the questionnaire. The language and education level of the respondents was a focus during the development of all questionnaire modules, and substantial efforts were made to simplify language and response options in all custom modules. Where appropriate, a colour bar, ranging from green to red, was added to help participants understand the value (positive/neutral/negative) of the response options (figure 3A). We also added a colour bar to each COPSOQ question, which helped to highlight that the response scales vary (positive to negative or negative to positive) by question (figure 3B).

To further assist employees, a workplace language course will be given 1 month prior to introducing the questionnaire for the 24 employees deemed to have

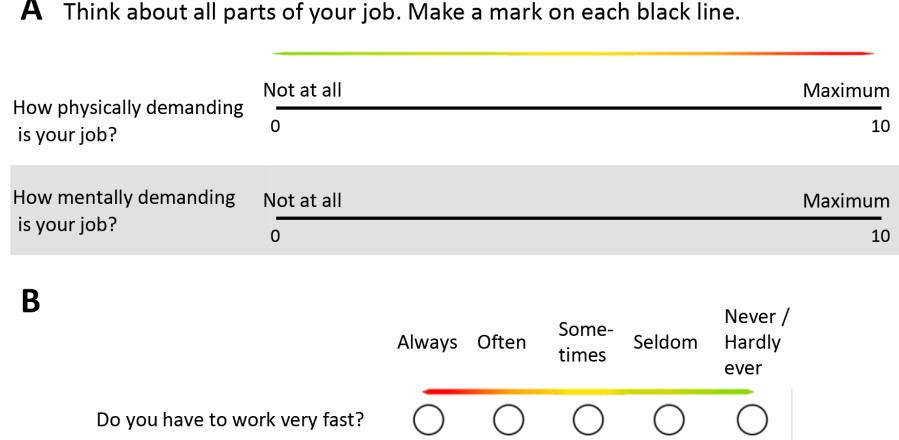

**Figure 3** Examples of survey questions showing colour bar used to assist in understanding the value of response options. (A) A custom module question and (B) A Copenhagen Psychosocial Questionnaire question.

the lowest language proficiency. As part of the course, a Swedish language teacher will review the questionnaire and teach employees how to respond to the different types of questions (short answer, multiple choice and CR-10 response scale). Lessons will be provided during working hours for groups of eight employees at a time, be 1 hour in duration, and occur two times a week. In addition, all employees will be provided with a copy of the questionnaire 1 week in advance of the data collection.

Questionnaires will be completed during paid working hours in small groups (6–8 employees) sitting in the same room as a researcher. We will provide help as needed and encourage workers to join a group with other workers sharing the same mother tongue. Mobile phones and translation/dictionary apps will be permitted and encouraged. Workers will also be permitted to answer the survey at home if desired. The on-site researcher will have trained in giving consistent explanations. The same researcher will be present on all occasions.

### Technical measurements

Pre implementation, technical data collection will be conducted to assess postures and HR over 5 consecutive days, and then to assess muscle activity levels during individual work tasks. All full-time permanent and contract employees working day shifts will be invited to participate in one or both technical data collections.

Post implementation, we will evaluate any work tasks that have changed and any new work tasks that have been introduced.

### Posture and HR assessment

Working postures, HR and HR variability will be assessed continuously over 5 consecutive working days. Postures will be assessed using wireless triaxial accelerometers (Axivity, Newcastle upon Tyne, UK) affixed to the skin using double-sided toupee tape (between the device and the skin) and breathable medical tape (over top of the device). Upper arm elevation will be recorded using units mounted according to 'LT INC' location[41]: upper edge of the device aligned with the insertion of the deltoid muscle into the humerus, long axis aligned with the humerus. Trunk elevation angle will be recorded using a unit placed on the sternum with the top edge of the unit at the level of the sternoclavicular notch, and the long axis of the unit aligned with the line of gravity. A final accelerometer unit will be placed on the right thigh (long axis aligned with the femur) and will be used to determine activities (sit/stand/walk/run) during the 5 working days.

During the same 5-day period, participants will also wear a BodyGuard HR monitor (Firstbeat Technologies Oy, Finland) comprising two electrodes (Ambu, Denmark) positioned according to Firstbeat recommendations, a wireless logger (attached to the electrode positioned directly below the right clavicle) and a wire joining the two electrodes.

To permit task-level analysis of posture and HR, workers will record the specific tasks and times at which each task was performed over the 5-day period in a daily logbook. A researcher will briefly meet each worker individually once a day during the 5-day period. Placement and alignment of all accelerometer units will be verified and tape changed, as necessary, depending on worker discomfort and tape adhesion, and calibration exercises will be performed according to Jackson et al.[41] The researcher will also confirm that the participant has been completing the work task log and responding to the fatigue assessment questions.

### Muscle activity assessment

Muscle activity will be assessed using electromyography (EMG) recorded continuously during one working bout of approximately 2 hours, and during the adjacent coffee break (15 min) or lunch break (30 min). During a work bout, workers typically perform one task, but may perform two or more tasks. EMG will be recorded bilaterally from the trapezius (Tr EMG) and forearm (FA EMG) muscles using a portable logger and EMG sensors from Biometrics (Newport, UK). At each recording site, the skin will be shaved and cleaned with alcohol prior to applying a disposable electrode (Ambu, Denmark). A single ground electrode (Ambu, Denmark) will be positioned atop a prominent thoracic vertebra (usually C7). Tr EMG electrodes will be positioned 2 cm lateral to the midpoint between C7 and the acromion process and aligned along the direction of the muscle fibres.[42] Bilaterally, a 'through the forearm' assessment of FA EMG will be made, as described by Takala and Toivonen,[43] using electrodes placed just lateral to the muscle bellies of the flexor digitorum superficialis and extensor digitorum communis muscles. Given the physical nature of the laundry handling work tasks, the tight spaces through which workers are often required to navigate, and the heavy machinery involved in the work, care will be made to ensure the EMG recording equipment is securely attached to the participant and that the cables will not interfere with their work.

A rest file and three repeats of muscle-specific reference contractions will be collected to permit normalisation. For the trapezius muscles, a submaximal reference voluntary exertion (RVE) normalisation will be employed according to Mathiassen et al.[42] RVE trials will be 15 s in duration[44 45] interspaced by 30 s of rest. For the forearm, a maximum voluntary exertion (MVE) pinch grip normalisation task will be employed. Subjects will gradually increase their pinch grip on a pinch dynamometer over 3 s, until reaching and holding their MVE for 3 s before gently ramping down their exertion. MVE trials will be approximately 7 s in duration interspaced by 60 s of rest

A researcher will follow workers during the EMG recording period to document the times and tasks performed. Workers will also be videotaped during the EMG recording period to provide additional assistance in dividing the full EMG recording into component tasks.

### Focus group interviews

Semistructured focus group interviews will be conducted with each job rotation team and with individual managers at three time points and analysed as outlined in the 'Implementation process and outcome evaluation' section above. Information regarding facilitators and barriers to the intervention and additional changes occurring during the follow-up period will be used to give additional context to the survey and technical measurement data.

### Company data

The laundromat will track and provide weekly data on production rate (total tonnes laundered), quality, worker sick days, days off for workers to care for sick children, accidents, injuries and worker turnover from baseline until the end of the follow-up period.

### Data processing and statistical analysis

Most previous studies documenting working conditions and occupational physical exposures via questionnaires and technical measurements have data analyses based on standard statistical methods, including a priori power calculations. Such methods assume that participants represent, in theory, the total population of all workers in similar settings. In our study, results consider a specific workplace and its 60 participant workers and not all workers employed in all commercial laundromats. We accept this notion in the research project and acknowledge it by adjusting basic descriptive statistics, such as SDs, by a finite population correction term.[35] In terms of the finite population correction, the anticipated number of technical measurement participants corresponds to approximately 30% of the total worker population.

To estimate a study size necessary to obtain sufficient power for assessing significant differences requires information regarding the relevant and interesting effect sizes; however, these data are not currently available for most of our quantitative variables. Further, information regarding the variance of the variables is also required but, again, these data are not currently available. Accordingly, in this case study of a single laundromat, we will aim to document changes according to our pre–post design.

Differences pre to post job rotation in work task allocation, psychosocial work environment conditions (including COPSOQ dimension) and self-reported health will be analysed, including consideration of confounders such as age, geographical location, marital status, family situation and other socioeconomic factors ultimately included in the questionnaires. Change scores will be calculated between baseline and follow-up for all COPSOQ dimensions.

The extent to which factors differed between men and women and between Swedish-born and foreign-born employees will, to the extent possible, be considered using a stratification approach.

The Firstbeat SPORTS uploader (V.1.0; Firstbeat Technologies) will be used for uploading and visually

identified artefacts will be removed. Average HR during work, non-work and sleep will be calculated using a custom MATLAB script. Relative aerobic workload will be calculated in a previously developed custom MATLAB script and will be based on the HR reserve (HRR), which is a well-established estimate of the body's workload and the body's workload relative to individual fitness.

The accelerometry data will be processed using a custom-made MATLAB programme to determine the time spent sitting, standing, walking, in light physical activity and in moderate to vigorous physical activity during both working and non-working hours. Trunk and upper arm elevation data will be calibrated[41] to angles calculated using an in-house custom MATLAB programme.

Accelerometry and HR data will be determined for each individual work task using the log book kept by workers. For each task, HR variables calculated will include HR intensity (relative aerobic workload, measured in terms of % HRR), time at different intensity levels, and HR variability, and postural variables, mean angle, time in neutral and minute-to-minute variance of angle.[46] Tasks where worker HR reaches or exceeds 60% HRR will be deemed sufficiently high-intensity for potentially generating increases in physical capacity.[47]

Post collection, EMG signals will be Butterworth filtered (30 Hz high pass) for minimisation of ECG contamination [48], offset corrected and root mean square converted (100 ms moving window), then quadratically rest adjusted. All files will be visually inspected to ensure good data quality. The mean amplitude across a stable 10 s period from each of the three RVE trials will be calculated and the mean of the three trials will be used to normalise the subject's Tr-EMG data. The largest 0.5 s mean EMG value will be determined across each MVE, and the largest single MVE will be used to normalise the subject's FA-EMG

EMG data from the approximately 2-hour work bout will be partitioned, if necessary, to individual work tasks using the time log kept by the researcher during data collection and, when necessary, assisted by the time stamp in the video data. For each task, the following muscle activity variables will be calculated using a custom-made MATLAB programme: mean, peak, time at rest, minute-to-minute variance and sustained low-level muscle activity (SULMA) periods.[49]

A task-based exposure matrix will be constructed comprising the posture and muscle activity variables, as outlined above. Using the log book data, individual exposure profiles will be constructed by estimating exposure variation for each day using an approach similar to the job variance ratios described by Barbieri *et al*.[46] Weekly variation will be assessed at the individual level by the mean variation across days and the between-days variance across the 5 days.

Pre–post changes in individual estimates of daily and weekly exposure variation will be assessed using repeated measures analysis of Variance (ANOVA) analyses and, in cases where multiple variables reflect similar properties of exposure, using multivariate ANOVA (MANOVA)

analyses. ANOVA analyses will include a correction for the effects of multiple comparisons.

### Interview data

To the extent possible, differences in experiences of the job rotation will be considered between male and female employees and Swedish-born and foreign-born employees. Management interview data will be primarily used to assist in interpreting and understanding survey and technical data findings including changes occurring during the follow-up that could have influenced work organisation or work environment unrelated to the intervention.[33]

In interpreting focus group interviews, Acker's theory of inequality regimes will be used as a framework for assessing how inequalities are produced and reproduced.[50] Acker's theory comprises both organisational and individual levels and states that inequality regimes exist in all organisations and can be defined as 'loosely interrelated practices, processes, actions and meanings' that create power orders from intersections of gender and ethnicity within the organisation.[50]

### Patient and public involvement

No patient involved.

### Registration

The study is preregistered with the Open Science Framework (https://osf.io/zmdc8/).

## ETHICS AND DISSEMINATION

The study received approval from the Swedish Ethical Review Authority (reference number 2019-00228). Written informed consent will be obtained for each data collection method, namely: accelerometry and HR monitoring, EMG, survey and focus group interview participation.

The results of the project will be shared directly with the Elis employees and managers at the participating laundromat, the worker's union (IF Metall) and the central Swedish and Global Elis concerns via presentations and summary reports. Research results will also be disseminated via publication in international, peer-reviewed, open-access scientific journals and presentation at international scientific conferences.

### Strengths and Limitations

The major novel contributions of this study to the literature are (1) the broad scope of the evaluation of the job rotation intervention, including consideration of the effects of the rationalisation on work environment conditions, health and production quality and resilience with consideration to gender and social equality and (2) the inclusion of a process evaluation of the intervention implementation.

The consideration of inequality both from a male–female (gender) and a place of birth (ethnicity) perspective is in line with intersectionality theory[51] and further

reflects the reality of modern immigration. It also reflects the Swedish government's latest Work Environment Strategy, which prioritises increased efforts towards sustainability with the goal of all employees (regardless of sex or place of birth) having a work environment that makes them able to, capable of and willing to work until official retirement age.[52]

The primary limitation of this longitudinal case study is the limited number of workers, and the inclusion of a single work site, which will limit the generalisability of the results. Still, we anticipate that our findings on the impact of job rotation on physical and psychosocial work conditions, health and equality will be immediately useful for other laundromats in the global Elis concern. Further, we anticipate findings regarding social equality to be relevant for other commercial laundromats and small-sized and medium-sized production-based industries with high employee diversity. The study is also limited by the lack of information required to perform power calculations a priori (ie, knowledge of the relevant/interesting effect sizes and size of variance for most of the quantitative variables). On completion of this study, these data will be available to guide future studies.

**Acknowledgements** The authors wish to thank the production workers at the Elis laundromat in Ockelbo Sweden for generously welcoming the research team and embarking on this journey together with genuine enthusiasm.

**Contributors** JJ and SM conceptualised the project. All coauthors (JJ, MS, GBL, LM and SM) participated in the cocreation process in which the study design and intervention were developed with JJ leading the process. All authors made substantial contributions to the development of the research study aims. JJ, MS and GBL developed the job rotation. JJ and SM developed the research study design. JJ collected pilot data to assess task exposure variation. All authors approve the submitted version and agree to be personally accountable for their contributions. All authors read and approved the final manuscript.

**Funding** This work was supported by the Swedish Research Council for Health, Working Life and Welfare (Forte) grant number 2009-1761.

**Competing interests** None declared.

**Patient and public involvement** Patients and/or the public were involved in the design, or conduct, or reporting, or dissemination plans of this research. Refer to the Methods section for further details.

**Patient consent for publication** Not applicable.

**Provenance and peer review** Not commissioned; externally peer reviewed.

**ORCID iD**
Jennie A. Jackson http://orcid.org/0000-0003-2939-0236

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
