## [Reviewer comments · BMJ Open]

ARTICLE DETAILS

TITLE (PROVISIONAL)	Assessing the efficacy of a job rotation for improving occupational physical and psychosocial work environment, musculoskeletal health, social equality, production quality, and resilience at a commercial laundromat: Protocol for a longitudinal case study
AUTHORS	Jackson, Jennie; Sund, Marianne; Barlari Lobos, Griztko; Melin, Lars; Mathiassen, Svend

VERSION 1 – REVIEW

REVIEWER	Ordudari, Zahra Isfahan University of Medical Sciences
REVIEW RETURNED	11-Oct-2022

GENERAL COMMENTS	The reviewer provided a marked copy with additional comments. Please contact the publisher for full details.
--

REVIEWER	Liebers, Falk Federal Institute for Occupational Safety and Health (BAuA), Nöldnerstr. 40-42, 10317 Berlin, Germany, FB 3
REVIEW RETURNED	13-Jan-2023

GENERAL COMMENTS	Peer-Review for BMC Open Draft ID • bmjopen-2022-067633 Authors of the draft: • Not blinded to the reviewer: Jackson et al. Title of the draft: Assessing the efficacy of a job rotation for improving occupational physical and psychosocial work environment, musculoskeletal health, social equality, production quality, and resilience at a commercial laundromat: Protocol for a longitudinal case study. Version of the draft / release date: • First draft; 05 Jan 2023 (downloaded by the reviewer) Date of the review • 13 Jan 2023 Type of the manuscript • Study protocol Recommendation of the reviewer
---

	1 Generally aspects  • Thank you for the opportunity to review the interesting protocol. • The manuscript describes the design (not the results) of an ongoing interventional study (study protocol). • The study aims to develop a job rotation (intervention) with company stakeholders of a commercial laundromat (exposure), perform a process evaluation of the implementation, and determine the extent to which the intervention improves (outcomes) the physical and psychosocial work environment, indicators of health, gender and social equality among workers (population), and production quality and resilience. • Generally, the described study design refers to a non-blinded, non-randomized, but pragmatic interventional study in an occupational setting with process and outcome evaluation. Therefore the subject of the study protocol is very interesting and in the scope of BMC Open. Abstract, background and aim of the study, hypotheses, methods, and statistical analysis are understandable and well written. The study protocol is structured and it provides all necessary information. • Nevertheless, the reviewer would like to provide some comments and recommendations to the protocol. 2 Abstract, title, keywords  • The title is informative and reflects the main content and aim of the paper. • The abstract provides an informative and balanced summary about background, aim of the study, methods, approaches, and ethical aspects. • The keywords “repetitive work, variation, work task organization” are given. • Comment 2.1 “keywords” (recommendation): The keywords do not really reflect the main purpose of the study. Okay, the key words should not repeat terms of the title or abstract. But “job rotation”, “interventional study”, “process and outcome evaluation” etc. are missing. I would suggest to add such term to the keywords 3 Introduction/Background  • The introduction is informative and explains the rationale for the study. The authors provided the scientific background and cited more or less relevant studies in the field. The citation of the given scientific sources seems to be correct. The sources are current and relevant. The authors provide the aims of the study within the first paragraph of section “methods and analysis” (“... to develop a job rotation intervention at a commercial laundromat in concert with company stakeholders using co-created program logic and to determine the extent to which the intervention can improve the physical and psychosocial work environment and indicators of health, advance gender and social equality among workers, and increase production quality and resilience without negatively affecting production rates.”). • Comment #3.1 (page 6, line 47 to page 7, line 10 recommendation): The reviewer recommends to provide the aim of the study separately as section “aim of the study” or at the end of section “Introduction”. 4 Section “Methods”  • The section provides all relevant and expected information regarding such a interventional study. The study design, intervention, the study population and data sources, settings, and
--	--

	the specific characteristics of the different data collections (questionnaire, interviews, technical measurements, postural assessments, heart rate monitoring, emg measurement etc.) are mentioned and sophisticatedly described. The statistical approaches are provided.  • Comment #4.1 (page 10, line 46, section “Data processing and statistical analysis”, recommendation: The authors mentioned that a power calculation is not appropriate because the number of participants is given. The reviewer recommends the authors should provide the minimum effect we could detect considering the given number of participants based on knowledge about mean and variance or prevalence of the considered outcomes. I think it will also be helpful to think about the minimum relevant effect for each outcome, which the authors would considered or interpreted as success of the intervention. • Comment #4.2 (page 13, line 4, “ANOVA”, comment): The authors describe a very complex interventional study. But the authors provide the following statement: “... pre-post changes in individual estimates of daily and weekly exposure variation will be assessed using a repeated measures ANOVA.” The problem with ANOVA is that you have to run many statistical analyses to consider all outcomes. Please consider to use MANOVA or other multi-level approaches to include more as one outcome and to adjust for multiple testing. 5 Tables / Figures  • Three figure are included in the draft. The figures and the titles of the figures are understandable. • No comments, remarks or revisions. 6 Others aspects  • The ethical approval and information are provided. • Author’s contribution, acknowledgements, declaration regarding conflict of interest and information on funding are provided. • No comments, remarks or revisions. 7 Section References / Citations in the manuscript  • Citations of references are provided numbered within the draft. • The draft includes 49 references. The reference list is order of the citation within the manuscript. The citation of the references seems to be correct. • No comments, remarks or revisions.
--	---

REVIEWER	Descatha, Alexis UNIV Angers Inserm , U1085 Irset, Ester Unit
REVIEW RETURNED	22-Jan-2023

GENERAL COMMENTS	I read with a particular interest the protocol. However, I am totally lost since the format is far from what I am expecting, probably because I am a physician/epidemiologist specialist in occupational health. Indeed, I am expecting to find a precise aim, population well defined (how/ when and the dates as mentioned/ where) and precise number (not approximatively), variables, outcomes, and
---

	analysis plan, enough detailed to be reproduced. What is presented here is a protocol of potential intervention that the author would “co-create”, but we are expecting the protocol of the intervention and its evaluation.
--	--

REVIEWER	Sato, Tatiana de Oliveira Universidade Federal de São Carlos, Physical Therapy
REVIEW RETURNED	21-Feb-2023

GENERAL COMMENTS	This is a very interesting longitudinal case study on the implementation of job rotation in a laundromat. There are some comments for the authors:  1. Sample size numbers are not clear (e.g. in abstract n=60; in methods section n=50). Please clarify; 2. The evaluation of the implementation process is one of the most interesting parts of the study. I suggest that the authors describe it in more detail in the Methods section; 3. The definition of the job rotation scheme is also a critical point. Please, provide details about of how high and low demand tasks were defined and organized in the schema; 4. In the "Study Design Overview" include the planned dates; 5. Do the authors think that 15 participants in the technical measurements will be sufficient to provide evidence on the effects of job rotation? How will these 15 workers be selected?
---

REVIEWER	Karkkainen, Sanna Finnish Institute for Health and Welfare
REVIEW RETURNED	15-Mar-2023

GENERAL COMMENTS	This study protocol provides a comprehensive description of the planned research, including careful attention to potential risks, limitations, and ethical considerations. The plan includes collecting questionnaire information, technical measurement data and interview information. As a small note, collecting the company's perspective regarding the process could provide distinctive contextual insights and mitigate the impact of a limited sample size. Nonetheless, the current study protocol is robust and provides important information with the aim of improving work environment and reducing musculoskeletal complaints.
---

VERSION 1 – AUTHOR RESPONSE

Reviewer: 1

Dr. Zahra Ordudari, Isfahan University of Medical Sciences

Comments to the Author:

Dear authors, pay attention to the comments in the manuscript file.

The comments were copied from the PDF copy of the manuscript and are included alongside our responses below:

1 - Results and conclusions should be added to the abstract

Since this is a protocol article, we do not have results and conclusions.

2 - Use the past tense in the article

It is our understanding that past tense should be reserved for articles once a study has been completed, while future tense is appropriate for a protocol article describing a study that has not yet been completed. We have elected to continue using the future tense for describing the methods in this protocol paper.

3 – Keywords: Please use more appropriate words, such as work rotation, musculoskeletal health, etc.

The following keywords have now been added: work rotation, job rotation, musculoskeletal health, intervention, implementation, co-creation, co-created program logic, process and outcome evaluation

4 - The purpose of the study is usually mentioned at the end of the introduction

The general and specific aims have been moved to the end of the introduction. We also found it strange to place them in the methods section but originally did so as stated in the guidelines provided on the BMJ Open website.

5 – Methods Section - Technical measurement data: Please merge this section with the similar section in the procedure

We have revised the layout of the methods section to clearly delineate between the three parts of the study protocol and their respective data collection and analysis aspects. We hope this change will satisfy your request.

6 - The article does not have results, discussion and conclusions.

Since this is a protocol article, we do not yet have results or a discussion and conclusions about the results. These pieces will follow in subsequent articles once the data collection is completed.

Reviewer: 2

Dr. Falk Liebers, Federal Institute for Occupational Safety and Health (BAuA), Nöldnerstr. 40-42, 10317 Berlin, Germany

Comments to the Author:

Peer-Review for BMC Open

Draft ID

• bmjopen-2022-067633

Authors of the draft:

• Not blinded to the reviewer: Jackson et al.

Title of the draft:

Assessing the efficacy of a job rotation for improving occupational physical and psychosocial work environment, musculoskeletal health, social equality, production quality, and resilience at a commercial laundromat: Protocol for a longitudinal case study.

Version of the draft / release date:

• First draft; 05 Jan 2023 (downloaded by the reviewer)

Date of the review

• 13 Jan 2023

Type of the manuscript

• Study protocol

Recommendation of the reviewer

1 Generally aspects

• Thank you for the opportunity to review the interesting protocol.

- The manuscript describes the design (not the results) of an ongoing interventional study (study protocol).
- The study aims to develop a job rotation (intervention) with company stakeholders of a commercial laundromat (exposure), perform a process evaluation of the implementation, and determine the extent to which the intervention improves (outcomes) the physical and psychosocial work environment, indicators of health, gender and social equality among workers (population), and production quality and resilience.
- Generally, the described study design refers to a non-blinded, non-randomized, but pragmatic interventional study in an occupational setting with process and outcome evaluation. Therefore the subject of the study protocol is very interesting and in the scope of BMC Open. Abstract, background and aim of the study, hypotheses, methods, and statistical analysis are understandable and well written. The study protocol is structured and it provides all necessary information.

Thank you for the great summary and kind words.

- Nevertheless, the reviewer would like to provide some comments and recommendations to the protocol.

2 Abstract, title, keywords

- The title is informative and reflects the main content and aim of the paper.
 - The abstract provides an informative and balanced summary about background, aim of the study, methods, approaches, and ethical aspects.
 - The keywords “repetitive work, variation, work task organization” are given.
- Comment 2.1 “keywords” (recommendation): The keywords do not really reflect the main purpose of the study. Okay, the key words should not repeat terms of the title or abstract. But “job rotation”, “interventional study”, “process and outcome evaluation” etc. are missing. I would suggest to add such term to the keywords

Good point! The following keywords have now been added:

work rotation, job rotation, musculoskeletal health, intervention, implementation, co-creation, co-created program logic, process and outcome evaluation

3 Introduction/Background

- The introduction is informative and explains the rationale for the study. The authors provided the scientific background and cited more or less relevant studies in the field. The citation of the given scientific sources seems to be correct. The sources are current and relevant. The authors provide the aims of the study within the first paragraph of section “methods and analysis” (“... to develop a job rotation intervention at a commercial laundromat in concert with company stakeholders using co-created program logic and to determine the extent to which the intervention can improve the physical and psychosocial work environment and indicators of health, advance gender and social equality among workers, and increase production quality and resilience without negatively affecting production rates.”).
- Comment #3.1 (page 6, line 47 to page 7, line 10 recommendation): The reviewer recommends to provide the aim of the study separately as section “aim of the study” or at the end of section “Introduction”.
- The general and specific aims have been moved to the end of the introduction as per the request of 2 of the reviewers. We found it strange to place them in the methods section, but initially followed the guidelines provided on the BMJ Open website.

4 Section “Methods”

- The section provides all relevant and expected information regarding such a interventional study. The study design, intervention, the study population and data sources, settings, and the specific characteristics of the different data collections (questionnaire, interviews, technical measurements, postural assessments, heart rate monitoring, emg measurement etc.) are mentioned and sophisticatedly described. The statistical approaches are provided.

- Comment #4.1 (page 10, line 46, section “Data processing and statistical analysis”, recommendation: The authors mentioned that a power calculation is not appropriate because the number of participants is given. The reviewer recommends the authors should provide the minimum effect we could detect considering the given number of participants based on knowledge about mean and variance or prevalence of the considered outcomes. I think it will also be helpful to think about the minimum relevant effect for each outcome, which the authors would considered or interpreted as success of the intervention.

Revisions and additions were made to the methods section (page 11) as follows (new text shown in bold)::

Most previous studies documenting working conditions and occupational physical exposures via questionnaires and technical measurements have data analyses based on standard statistical methods, including a priori power calculations. Such methods assume that participants represent, in theory, the total population of all workers in similar settings. In our study, results consider a specific workplace and its 60 participant workers and not all workers employed in all commercial laundromats. We accept this notion in the research project and acknowledge it by adjusting basic descriptive statistics, such as SDs, by a finite population correction term (35). In terms of the finite population correction, the anticipated number of technical measurement participants corresponds to approximately 30% of the total worker population.

To estimate a study size necessary to obtain sufficient power for assessing significant differences requires information regarding the relevant and interesting effect sizes, however these data are not currently available for most of our quantitative variables. Further, information regarding the variance of the variables is also required but, again, these data are not currently available. Accordingly, in this case study of a single laundromat, we will aim to document pre-post changes descriptively.

- Comment #4.2 (page 13, line 4, “ANOVA”, comment): The authors describe a very complex interventional study. But the authors provide the following statement: “... pre-post changes in individual estimates of daily and weekly exposure variation will be assessed using a repeated measures ANOVA.” The problem with ANOVA is that you have to run many statistical analyses to consider all outcomes. Please consider to use MANOVA or other multi-level approaches to include more as one outcome and to adjust for multiple testing.

The methods section (Data processing and statistical analysis section) has been revised as follows (new text shown in bold)::

Pre-post changes in individual estimates of daily and weekly exposure variation will be assessed using repeated measures ANOVA analyses and, in cases where multiple variables reflect similar properties of exposure, using MANOVA analyses. ANOVA analyses will include a correction for the effects of multiple comparisons.

5 Tables / Figures

- Three figure are included in the draft. The figures and the titles of the figures are understandable.

- No comments, remarks or revisions.

6 Others aspects

- The ethical approval and information are provided.
- Author's contribution, acknowledgements, declaration regarding conflict of interest and information on funding are provided.

- No comments, remarks or revisions.

7 Section References / Citations in the manuscript

- Citations of references are provided numbered within the draft.
- The draft includes 49 references. The reference list is order of the citation within the manuscript. The citation of the references seems to be correct.

- No comments, remarks or revisions.

Reviewer: 3

Prof. Alexis Descatha, UNIV Angers Inserm , CHU Angers Pôle A Vasculaire

Comments to the Author:

I read with a particular interest the protocol. However, I am totally lost since the format is far from what I am expecting, probably because I am a physician/epidemiologist specialist in occupational health. Indeed, I am expecting to find a precise aim, population well defined (how/ when and the dates as mentioned/ where) and precise number (not approximatively), variables, outcomes, and analysis plan, enough detailed to be reproduced. What is presented here is a protocol of potential intervention that the author would "co-create", but we are expecting the protocol of the intervention and its evaluation.

The general and specific aims were previously located at the start of the Methods and Analysis section, as per the author instructions for Protocol Articles on the BMJ Open website, but they have now been moved to the end of the introduction. We hope this will make them easier to find.

The population is defined as precisely as possible but given that this is a protocol article describing a study for which the data collection is not yet finished, we are unable to provide exact numbers. We have described the number of participants which we intend to include:

'For the questionnaire, we will aim for a response rate of 70% (n = 35), and we will aim for 15 participants in both the technical measurement protocols.' – Methods section, study population paragraph.

Additional detail regarding the required number of participants to detect a minimum effect has been added to the Methods and Analysis- Data processing and statistical analysis section – see page 11.

The study comprises three pieces, each with its own set of methods. The pieces include: (i) development of an innovative job rotation intervention via co-creation with company stakeholders, (ii) evaluation of the implementation of the job rotation intervention, and (iii) assessment of the efficacy of the job rotation intervention using a pre-post intervention design. We have re-structured the methods section to group the methods more clearly into their respective piece of the study.

The co-creation approach to developing interventions in conjunction with company stakeholders was employed to attempt to obtain a maximum impact of the job rotation intervention by matching the needs and constraints of the company. The process is outlined in the first paragraph of the Methods

and Analysis section, 'Development of the job rotation intervention using co-created program logic'. Describing this process in more detail is one of the aims of the study, and thus additional information is considered a result that will ultimately stem from this study.

The co-creation approach was also used to ensure that company and researcher stakeholders agreed upon the aims of the study, the set of variables that were collected, and the timeline and procedure for the study to evaluate the impact of the job-rotation implementation using the pre-post implementation design, as outlined in the text.

Regarding the variables and outcomes that characterize the changes in work environment and health related to the job rotation intervention, we believe these variables are thoroughly described in the Methods and Analysis section. The proposed analysis is also outlined in that section. Several revisions have been made to the analysis to offer more detail regarding the analysis, as per the specific request of other reviewers (see red text in attached manuscript), which may also help with your concerns.

Reviewer: 4

Dr. Tatiana de Oliveira Sato, Universidade Federal de São Carlos

Comments to the Author:

This is a very interesting longitudinal case study on the implementation of job rotation in a laundromat. There are some comments for the authors:

1. Sample size numbers are not clear (e.g. in abstract $n=60$; in methods section $n=50$). Please clarify; The methods text has been updated to the correct number ($n = 60$).

2. The evaluation of the implementation process is one of the most interesting parts of the study. I suggest that the authors describe it in more detail in the Methods section;
Good suggestion! We agree it wasn't as clearly described as it should be. We have added an intro sentence at the start of the methods section and re-structured the methods to clearly reflect the three aspects of the study protocol, namely: (i) development of an innovative job rotation intervention via co-creation with company stakeholders, (ii) evaluation of the implementation of the job rotation intervention, and (iii) assessment of the efficacy of the job rotation intervention using a pre-post intervention design.

All methods regarding the implementation evaluation have now been grouped in a new section in the methods entitled, 'Implementation process and outcome evaluation'. Additional text regarding the evaluation analysis was also added to this section.

3. The definition of the job rotation scheme is also a critical point. Please, provide details about of how high and low demand tasks were defined and organized in the schema;

The specific tasks and the way they were defined and organized are key to the job rotation that we developed, and we believe that this is a result of the first aspect of the study outlined in our protocol paper (i.e. Development of the job rotation intervention using co-created program logic – cf new methods section 1). Accordingly, we feel like these specific data should be reserved for the paper we plan to publish outlining the co-creation program logic process used to develop the intervention and study design, as per aim 1 of the current paper.

4. In the "Study Design Overview" include the planned dates;
We do not know the exact dates so are unable to include them.

5. Do the authors think that 15 participants in the technical measurements will be sufficient to provide evidence on the effects of job rotation? How will these 15 workers be selected?

The following paragraph has been added to the methods section, and we will aim for 20 workers:

In our study, results consider a specific workplace and its 60 participant workers and not all workers employed in all commercial laundromats. We accept this notion in the research project and acknowledge it by adjusting basic descriptive statistics, such as SDs, by a finite population correction term (35). In terms of the finite population correction, the anticipated number of technical measurement participants corresponds to approximately 30% of the total worker population. To estimate a study size necessary to obtain sufficient power for assessing significant differences requires information regarding the relevant and interesting effect sizes, however these data are not currently available for most of our quantitative variables. Further, information regarding the variance of the variables is also required but, again, these data are not currently available. Accordingly, in this case study of a single laundromat, we will aim to document changes according to our pre-post design.

All workers will be invited to participate, and barring a response greater than 50% (ie $n = 30$), which from previous experience in conducting many field studies with technical measurements is highly unlikely, we will measure all interested workers. This information has been added to the methods, Study population section (page 8):

For both technical measurement protocols we will collect data from all interested participants, and will aim for at least 20 participants.

Reviewer: 5

Dr. Sanna Karkkainen, Finnish Institute for Health and Welfare

Comments to the Author:

This study protocol provides a comprehensive description of the planned research, including careful attention to potential risks, limitations, and ethical considerations. The plan includes collecting questionnaire information, technical measurement data and interview information.

As a small note, collecting the company's perspective regarding the process could provide distinctive contextual insights and mitigate the impact of a limited sample size. Nonetheless, the current study protocol is robust and provides important information with the aim of improving work environment and reducing musculoskeletal complaints.

Thank you for the comment and encouraging words. We agree that the company's perspective regarding the process is important, and we collected this data by interviewing managers throughout the follow-up period but had not clearly described it in the submitted protocol paper.

We have now re-structured the methods section to more clearly differentiate between the three aspects included in the protocol article, namely: (i) development of an innovative job rotation intervention via co-creation with company stakeholders, (ii) evaluation of the implementation of the job rotation intervention, and (iii) assessment of the efficacy of the job rotation intervention using a pre-post intervention design. We have added additional information regarding the data we collected from company management in the new methods section entitled: Implementation process and outcome evaluation as follows (new text shown in bold):

To assess the implementation process, semi-structured focus group interviews with each job rotation team and individual interviews with management will be conducted at three time points: immediately prior to the start of the job rotation intervention and at 6 weeks and 12 months follow-up. Data collected immediately prior to the job rotation intervention will be used to assess worker readiness for change and worker impressions of their new working schedules. Individual interviews with managers from this time point will be used to assess readiness to lead change. At 6-weeks follow-up, group interviews will be used to document the extent to which the job rotation intervention has been implemented during the initial phase and self-reported ability of workers to complete all assigned tasks. At 12 months follow-up, group interviews with each team will be used to document the progress and extent of implementation achieved. Individual interviews with management will be used to assess the company's perspective and experience of the implementation process and the extent of success. Together, the interviews will capture facilitators and barriers to the implementation process and will document additional changes occurring during the follow-up period that were un-related to the intervention, but which may have affected the way in which work was performed. Interview data will be transcribed and analysed thematically (32), guided by research aim 2 to assess readiness for change and the extent of intervention implementation.